# Spatiotemporal Variation of Urban Plant Diversity and above Ground Biomass in Haikou, China

**DOI:** 10.3390/biology11121824

**Published:** 2022-12-14

**Authors:** Hai-Li Zhang, Mir Muhammad Nizamani, Josep Padullés Cubino, Lin-Yuan Guo, Jing-Jiang Zhou, Hua-Feng Wang

**Affiliations:** 1Hainan Yazhou Bay Seed Laboratory, Sanya Nanfan Research Institute, Hainan University, Sanya 572025, China; 2Key Laboratory for Sustainable Utilization of Tropical Bioresources, School of Life Sciences, Hainan University, Haikou 570228, China; 3Department of Plant Pathology, Agricultural College, Guizhou University, Guiyang 550001, China; 4Centre for Ecological Research and Forestry Applications (CREAF), 08193 Cerdanyola del Vallès, Spain; 5College of Plant Protection, Gansu Agricultural University, Lanzhou 730070, China; 6State Key Laboratory Breeding Base of Green Pesticide and Agricultural Bioengineering, Guizhou University, Huaxi District, Guiyang 550025, China

**Keywords:** plant diversity, Hainan, coastal tropical areas, above ground biomass, time-lag

## Abstract

**Simple Summary:**

The relationship between urban plant diversity (UPD) and above ground biomass (AGB) were assessed across 190 urban functional units in Haikou’s tropical urban ecosystems. UPD and AGB showed time-lag effects to human activities and socioeconomics.

**Abstract:**

Understanding the drivers of urban plant diversity (UPD) and above ground biomass (AGB) in urbanized areas is critical for urban ecosystem services and biodiversity protection. The relationships between UPD and AGB have been investigated simultaneously. However, the drivers of UPD and AGB have been explored independently in tropical coastal areas at different time points. To fill this gap, we conducted a remote sensing interpretation, field plant plot surveys, and compiled socioeconomic and urban greening management survey data. We conducted spatial analyses to investigate the relationships among UPD and socioeconomic variables across different primary and secondary urban functional units (UFUs) in the tropical urban ecosystems of the coastal city of Haikou, China. The primary UFUs with the highest AGB were the recreation and leisure districts in 2015 and 2021. In 2015, AGB was mainly correlated with the number of herb species in undeveloped land and the districts of industry, business, recreation, and leisure. In 2021, AGB was affected primarily by the frequency of fertilizing, maintenance, and watering. Our study found that the relationship between UPD and AGB varied across time and space in Haikou. The plant diversity and AGB’s response to human activities and socioeconomics appear to have a time-lag effect. These results provide new insights in understanding how management decisions affect urban vegetation and could be used to guide future urban green space planning in Haikou.

## 1. Introduction

Urban plant diversity (UPD) provides numerous ecosystem services that support the well-being of urban residents, such as recreation, education, biodiversity conservation, and the healthy development of urban ecosystems [1,2]. A large proportion of urban vegetation is usually purchased from nurseries and planted in urban green spaces [3,4]. Therefore, urban vegetation is usually limited by resources and social factors, such as the time and money spent on purchasing and growing plants [5,6]. Meanwhile, there is a close correlation between UPD and above ground biomass (AGB) [2,5]. AGB is associated with many ecosystem services, and as such, it has a good relationship with biodiversity in natural systems [6]. However, the relationship of AGB with plant biodiversity in urban areas remains poorly understood [7]. Since urban plant diversity and AGB distributions tend to vary with different spatiotemporal effects, no general conclusions have been drawn. A better understanding of the drivers of UPD and AGB may help to improve the management and conservation of biodiversity in urban areas worldwide.

Few previous studies have attempted to investigate the relationships among UPD, AGB, and the societal drivers across different urban functional units (UFUs, i.e., the work units in cities) in tropical urban ecosystems [8]. UFUs are the basic components of a city and strongly influenced by the city’s history and urban development patterns [4,9]. Different UFUs represent different types of land use and can be classified into commercial areas, residential areas, commercial areas, government agencies, research institutes, hospitals, industry sites, hotels, parks, urban villages, farmlands, wastelands, etc. Due to the spatial heterogeneity of urban ecosystems, the responses of UPD to influencing factors have different spatial patterns and time-lag effects that can also alter their composition [10,11]. Jong et al. (2013) analyzed the lag effect of vegetation in the past 30 years and found that climate could explain about 54% of vegetation variations [12]. The analysis of the lag effect of vegetation in Spanish coastal urban areas over the past 50 years found that urbanization was associated with a substantial increase in vegetation coverage [13]. In urban natural forests, UPD typically increases rapidly in early stages and declines in later stages [14]. In managed urban forests, UPDs are primarily affected by human disturbance and management practices. These factors play important roles in regulating ecosystem services and creating heterogeneous forest structures [15]. Although UPD in forest ecosystems has been extensively explored, evidence for this functional link is still lacking in urban green spaces under different management practices [16].

To better understand the driving mechanisms and spatiotemporal variations of UPD and AGB in coastal urbanized areas, we utilized the tropical coastal city of Haikou (China) as a study area and adopted the stratified random sampling method to set up 190 survey plots in different UFUs. Our study simultaneously considers muti-factors on UPD and plant distribution. We combined remote sensing interpretation with field surveys to obtain the 2015 and 2021 UPD data in Haikou. Using socioeconomic data of different time periods, the spatial distribution pattern of plant diversity in different types of UFUs was analyzed. The correlations between plant diversity (number of tree, shrub, and herb species) and land use, socioeconomics, and management measure were elucidated. This study aims to elucidate (1) the distribution patterns of UPD in Haikou in 2015 and 2021; (2) analyze the variations and driving factors of UPD and AGB in Haikou between 2015 and 2021; and (3) analyze whether time-lag effects could explain UPD and AGB in Haikou.

## 2. Materials and Methods

### 2.1. Study Area

Haikou (E110°10′~1110°41′, N19°32′~20°05′) is the capital, and most populous city of China’s Hainan Province [17]. The total area of the study area is about 3145.93 km^2^. The city population in 2020 was 2,360,100 (Figure 1) [18]. We chose a built-up area of Haikou to make our study area representative of the typical urban landscape. The built-up area of Haikou is bordered by Qiongzhou Strait on the north, the Haikou Ring Expressway on the south, the Haikou Nandu River of Haikou on the east, and Huoshankou Road on the west.

### 2.2. Selection of Urban Functional Units (UFUs) and Field Surveys

A total of 190 urban functional units (UFUs) were selected according to the study delineated by Zhang et al. (2022) [7]. The UFUs were divided into six primary types (public affairs service districts, industry districts, business districts, residential districts, recreation and leisure districts, transportation, and undeveloped land) and 16 secondary types (Table 1). The wetlands included marine areas and river channels that did not grow plants.

We used the stratified sampling to delineate plots in Haikou. We drew 154 grids of 650 × 650 m in the built-up area of Haikou. Within each grid, we selected 1–3 UFUs based on their representability and cultural importance [6,8]. A total of 190 UFUs were generated. Three plots of 20 × 20 m with high plant diversity in each UFU were selected to maximize the plant diversity of the entire UFU. Then, five 5 × 5 m shrub plots and five 1 × 1 m herb plots were established in the central four corners of each 20 × 20 m plot [2,19,20,21]. In all plots, we described and recorded the characteristics of all plants (species name, plant height and DBH for each tree; species name, plant height, clump number, and coverage for shrub species; species name, plant height, and coverage for herb species). We also used GPS to determine its longitude and latitude.

We conducted surveys in the same UFUs in May-June and November-December of 2015 and 2021, to ensure that the survey covered most of the flowering and fruiting periods. In a preliminary field survey in June and December, we found about 70% of the plants were flowering or bearing fruit. We used the “flora of China” to identify plant species and standardized plant names according to The Plant List (http://www.theplantlist.org/ (accessed on 15 June 2021)) [22]. If plant species could not be identified in the field, we collected samples to compare them with those in the China Flora Herbarium (https://www.cfh.ac.cn/ (accessed on 13 June 2021)) or consulted experts for identification. We distinguished between natural and cultivated plant species based on the definitions of Cheng et al. (2021) [23]. Natural species were those that grow without human disturbance, while cultivated plants were those that require human maintenance to grow and maintain viable populations.

### 2.3. AGB and UPD Calculations

We calculated AGB for each plant species with a diameter at breast height (DBH) ≥ 2 cm in each sampled plot using a formula proposed by Nizamani (2021):$$\mathrm{AGB}=0.4\pi {\left(\frac{\mathrm{dbh}}{2}\right)}^{2}\times \left(\mathrm{height}+300\right)\times \left(\mathrm{plant}\;\mathrm{density}\right)$$

The DBH data were obtained from field surveys, and the wood density was obtained from the trait vegetation database TRY (https://www.try-db.org/TryWeb/Home.php (accessed on 10 July 2021)). We used species richness as a measure of urban plant diversity (UPD), and calculated UPD separately for tree, shrub, and herb species in each UFU in 2015 and 2021.

### 2.4. Urban Management Variables

Management factors in our research included fertilization frequency (times/year), maintenance frequency (times/year), and watering frequency (times/year). We compiled a questionnaire and interviewed at least one leader or employee responsible for green maintenance at each UFU. If the managers were not present during the survey, we called the property management office to conduct a telephone survey to obtain the management data for the management factors.

### 2.5. Socioeconomic Variables

For housing prices, we obtained 2015 and 2021 data from Anjuke (https://haikou.anjuke.com (accessed on 1 July 2021)), a major online real estate software in China. The query time was July 2015 and July 2021. If the UFU was not residential, we used the average house price in the nearest residential area. For the construction age of the UFU, we inquired the property through the survey, or consulted the Baidu (https://www.baidu.com/ (accessed on 18 July 2021)) website to get the construction age of the building.

To determine the total population (P) of each UFU, we use the following formula: P = A × B × C × D; where A is the number of residential buildings in a UFU based on aerial photography and field investigation statistics, B is the number of floors in each building, C is the number of residential units (households) per floor, and D is the average number of people in each household (2.62 persons/family) according to 2015 and 2020 China National Census Data [24]. Finally, we calculated the population density (people/km^2^) as P/E, where E is the area covered by each UFU. We obtained the longitude and latitude of each UFU by GPS positioning.

### 2.6. Statistical Analysis

To compare UPD of different types of primary and secondary UFUs, we drew boxplots with AGB and the number of trees, shrubs, herbs, and total UPD in each UFU with the R package “ggplot2” [25]. We calculated z-scores for all variables using IBM SPSS Statistics 23.0 and removed outliers (z-score greater and lower than 3 and −3, respectively) [26]. The mean and standard deviation of AGB and the number of trees, shrubs, herbs and total UPD in each UFU were calculated using IBM SPSS Statistics 23 Chicago, USA. Then, we used AGB and UPD of trees, shrubs, and herbs species as dependent variable, and socioeconomic (construction age, housing price, population density), geographical factors (longitude, latitude,) and management factors (maintenance frequency, watering frequency, fertilization frequency) as independent variables. We used stepwise regression analysis and Akaike Information Criterion (AIC) to output the best model information. We first built two models using data from 2015 and 2021. Then, we built a third model to determine if UPD and AGB responses to socioeconomic and management factors had a time lag effect by comparing whether past socioeconomic factors better explain UPD than present socioeconomic factors. We built this model using UPD and AGB data from 2021 and socioeconomic and management factors from 2015.

## 3. Results

### 3.1. Urban Functional Unit Species Composition

According to our field surveys, we found 667 plant species belonging to 99 families in 2015 and 898 plant species belonging to 151 families in 2021. The families that contained the most plant species were Poaceae (N = 26), Asteraceae (N = 27), and Fabaceae (N = 32) in 2015, and Poaceae (N = 82), Fabaceae (N = 65), Asteraceae (N = 48) in 2021 (Appendix A).

### 3.2. Variations in AGB and UPD between 2015 and 2021 across Primary and Secondary UFUs

From 2015 to 2021 (Table 2 and Appendix A), the highest AGB increase was in public affairs service districts (2.78 × 10^7^ ± 3.27 × 10^7^) and the highest AGB decrease was in recreation and leisure districts (−5.31 × 10^7^ ± −3.19 × 10^7^). The top three primary UFUs with the greatest AGB were recreation and leisure districts (1.01 × 10^8^ ± 7.80 × 10^7^), industry and business districts (2.83 × 10^7^ ± 2.68 × 10^7^), and transportation (2.12 × 10^7^ ± 1.60 × 10^7^) in 2015 (Table 2; Figure 2), and recreation and leisure districts (4.8 × 10^7^ ± 4.88 × 10^7^), public affairs service districts (4.79 × 10^7^ ± 4.6 × 10^7^), and transportation (4.43 × 10^7^ ± 6.05 × 10^7^) in 2021.

From 2015 to 2021, the number of trees increased the most in transportation (2.80 ± 1.02), and the least in residential districts (0.90 ± 0.45). In 2015, the top three primary UFUs with the greatest number of trees were residential districts (4.38 ± 2.62), public affairs service districts (3.91 ± 1.95), and recreation and leisure districts (3.61 ± 1.52) (Appendix A; Figure 2). In 2021, these UFUs were transportation (6.13 ± 2.80), public affairs service districts (5.90 ± 2.84), and industry and business districts (5.65 ± 2.46).

From 2015 to 2021, the number of shrubs increased the most in undeveloped land (5.50 ± 0.70) and the least in recreation and leisure districts (2.78 ± 1.65). In 2015, the top three primary UFUs with the greatest number of shrubs were recreation and leisure districts (2.12 ± 1.46), residential districts (1.48 ± 0.55), and public affairs service districts (1.43 ± 0.58). In 2021, they were transportation (6.08 ± 2.90), industry and business districts (5.73 ± 2.03), and public affairs service districts (5.64 ± 2.36).

From 2015 to 2021, the number of herbs increased the most in transportation (6.90 ± 3.93) and decreased the most in recreation and leisure districts (−1.00 ± 1.50). In 2015, the top three primary UFUs with greatest number of herbs were recreation and leisure districts (4.48 ± 2.96), public affairs service districts (2.48 ± 1.79), and residential districts (2.02 ± 1.16). In 2021, they changed to transportation (8.04 ± 4.80), public affairs service districts (7.92 ± 4.98), and industry and business districts (7.35 ± 3.87).

From 2015 to 2021, the number of total species increased the most in transportation (22.5 ± 4.5). In 2015, the top three primary UFUs with greatest number of total species were recreation and leisure districts (10.21 ± 4.02), while in 2021 were transportation (22.5 ± 4.5).

Variations in AGB and UPD of the secondary UFUs between 2015 and 2021 are shown in Appendix B, Appendix A.

### 3.3. Models Predicting AGB in 2015 and 2021

AGB in 2015 was positively related to latitude (β coefficient = 0.008 **) (Table 3). AGB in 2021 had no significant relationship with construction age, population density, traffic flow, distance from main road, longitude, latitude, fertilizing frequency, maintenance frequency, or watering frequency. We found that species numbers in 2021 were affected by house prices in 2015 (Figure 3).

### 3.4. Models Predicting the Number of total Trees, Shrubs, and Herbs from Different Variables in 2015 and 2021

The number of tree species in 2021 was positively related to fertilization frequency (β coefficient = 0.276 *) (Table 4). The number of shrub species in 2015 had positive relationships with traffic flow (β coefficient = 0.133 *) and watering frequency ( β coefficient = 0.524 *). The number of shrub species in 2021 had a strong positive relationship with construction age (β coefficient = 0.360 ***) and distance from the main road (β coefficient = 0.090 *). The number of herb species in 2021 negatively correlated with fertilization frequency (β coefficient = −0.256 *). The total number of all species in 2015 had a positive relationship with watering frequency (β coefficient = 0.417 ***). The total number of all species in 2021 was positively related with maintenance frequency (β coefficient = 1.494 **).

## 4. Discussion

### 4.1. Variations of Species Composition

Most of the plant species found in Haikou’s UFUs were herbs. Despite of the differences in survey plot selection, our conclusions are similar to Guo et al. (2022): herbs are causal species that establish easily in new plots after dispersal by different vectors [27]. We found a large increase in the number of species in industry areas and sand land. Some plant species should be selected and planted in industrial areas due to their ability to tolerate pollutants and are used as pollution indicators, enrichment of pollutants (such as heavy metals) [28], dust prevention [29], and noise reduction [30]. Air pollution is a major problem arising mainly from industrialization. Plants remove a significant amount of pollution from the atmosphere as part of their normal functioning, thus directly increase the air quality in the city and its surrounding area. We should consider an integral part of any comprehensive plan aimed at improving overall air quality [31]. Urban sprawl has led to the exploitation of sand and wasteland, but this is not necessarily had a negative impact, because as shown by Zhang et al. (2022), the suburban areas of Haikou have become greener with urban expansion [7].

### 4.2. Variations in UPD and AGB in Primary and Secondary UFUs in 2015 and 2021

Different land use types affect the planting of trees, shrubs, and herbs in urban functional units, thus affecting AGB. Areas with more herbaceous species tend to have more biodiversity and higher biomass [32] (Zhou et al., 2017). Land use has widely been used to explain the distribution of urban plants. In 2015, our AGB was significantly positively correlated with most land use types in Haikou. These findings support the land use hypothesis, as different land types correspond to different government plans and impact on vegetation planting [9,33]. We found that the primary UFUs with the highest AGB in both 2015 and 2021 were recreation and leisure districts. The secondary UFUs with the highest AGB in 2015 and 2021 were parks. Like the findings of Goertzen et al. (2021) and Zhang et al. (2022), we found that recreation and leisure districts that usually include parks and other areas with large urban green spaces also had a high AGB [7,34]. The data on the change in AGB showed that AGB grew or remained largely unchanged between 2015 and 2021 because trees grow lusher with age, and UFU had hardly been torn down and rebuilt during the study.

The number of tree, shrub, and herb species was rising in each type of primary UFUs. The largest increase in the number of tree species was in the industry and business districts and transportation. The largest increase in the number of shrub species was in the undeveloped area, and the largest increase in the number of herb species was in transportation (Table 2). There were differences in the UPD within different UFUs, and we believe these differences are caused by people’s growing awareness that plants will improve urban environment, and because of people preferences for their own plant cultivation.

The species diversity of the park was relatively high in secondary UFUs. In contrast, the species diversity in government agencies and industrial areas had grown dramatically, and urban expansion led to increased plant diversity in sandy land between 2015 and 2021. Parks were large green patches in the city, which must be given the function of viewing, recreation, and protecting the ecological environment at the beginning of planning. Patch size is an important factor affecting UPD [35,36]. Previous studies showed that large forest patches were found to host many plant species that were not found in small patches, including many old-growth forest and rare species [37,38,39], and that remnant native vegetation was essential for the survival of native species [40,41]. We found that some parks in Haikou, such as Baishamen Park and West Coast Ribbon Park, were established on a desert beach where the vegetation was all secondary. However, they were also essential for the survival of native species.

The districts with the highest number of tree species in 2015 were residential districts in 2015 and public affairs service districts in 2021 [5]. Residential districts have large populations and tend to purchase large numbers of tree species for planting due to people’s preferences and the importance of urban biodiversity as the government has realized [35]. The Haikou government invested a lot of money in increasing urban green plants after 2014. More leisure space for urban residents was provided, thus making the 2021 public affairs service districts regional and increasing labor [15]. Previous research found that parks, universities, and government agencies usually own a large amount of land for grassland construction, such as government agencies and schools, gardens, and sports grounds. In 2021, the greatest number of herb species in Haikou City were government agencies. Using remote sensing images, we observed that during the period from 2015 to 2021, a large part of the area of government agencies was available for planting trees and building lawns to increase green areas.

### 4.3. Comparisons of the Drivers of AGB and UPD in Haikou in 2015 and 2021

In 2015 and 2021, the number of trees, shrub, and herb species were positively correlated with fertilizing frequency, maintenance frequency, and watering frequency. These results indicate that the increase in management was related to the species diversity within UFUs. Our research found that the AGB of different UFUs in Haikou in 2021 was significantly higher than in 2015. This may be because of the government policies for the urban green space in Haikou to gradually increase the green space in the city from 2004 to 2020 [42]. Therefore, these areas are likely to be used to enforce these policies. In 2015, the land use type was the most significant predictor of AGB, while AGB in 2021 was mainly affected by urban management methods. This conclusion further verifies that Haikou Municipal Government’s plan to increase urban green space was gradually realized. Our study found that plant diversity and AGB’s response to human activities and socioeconomics appear to have a time-lag effect. The time-lag effect of vegetation on climate change is an important aspect for explaining plant diversity [10,43].

Our model shows that socioeconomic factors in 2015 could partially explain AGB and shrub and herb species richness in 2021. In particular, the house prices in 2015 could explain AGB in 2021 better than the house prices in 2021, and the construction age in 2015 could explain shrub and herb species richness better than the construction age in 2021. The following reasons can explain this result. The luxury effect predicts a positive relationship between affluence and urban plant diversity. UFUs are ideal models for analyzing the components of urban vegetation based on the assumption that greater wealth allows greater investment in diversity [4,7,44]. Through field investigations, we found that the green space in the same UFU had rarely been torn down and rebuilt in these six years, and the plants had grown during these six years. This phenomenon appears widespread but lacks data support, so this view should be analyzed in future studies. At the same time, it enlightens us that the planning of green spaces should be far-sighted, and when investing in green spaces, we should estimate the landscape and ecological effects in the coming years.

Maintenance and watering frequency influenced urban plant diversity. The maintainers of UFUs will try to mitigate climate extremes by adjusting watering behavior to support plant survival especially in hot and dry weather [45]. Monitoring of watering frequency in community gardens in California, USA, found that when temperatures were higher, watering time was longer, and water consumption was higher [46]. This could demonstrate that water use behavior may be able to increase as well as maintain plant diversity. These results are consistent with our findings that in 2015 the total number of species was mainly affected by watering frequency. In 2021, the total number of species was mainly affected by maintenance frequency.

## 5. Conclusions

This study investigated the spatial and temporal variation of plant diversity and above ground biomass in Haikou across 190 UFUs over six years, which may only be the tip of the iceberg of our efforts to reveal the relationships among intricate factors in urban ecosystems. Through investigating Haikou’s urban spatial pattern and plant diversity, the choice of space and vegetation among UFUs should be differentiated by socioeconomic factors. Unlike previous random sampling, based on the combination of survey and re-mote sensing data, we found that within the urban functional unit of Haikou City from 2015 to 2021, land use and conservation management are the main drivers of AGB. Our findings suggest that recreation and leisure are essential for green space conservation in Haikou city. In particular, the government’s investment and effective application of the government’s financial investment in the greening of traffic areas has the most obvious effect, leading to the growth of a large amount of vegetation in a short period. In conclusion, our findings suggest that more governmental, commercial, and individual efforts are required to promote AGB in Haikou, and we advocate further research to understand the distribution of AGB at the UFU level. Moreover, we are conducting long-term monitoring of Haikou City as a case of the tropical coastal urban ecosystem. More scientific hypotheses (such as the time lag effect hypothesis) would be tested on larger spatial and temporal scales in the future.

## Figures and Tables

**Figure 1 biology-11-01824-f001:**
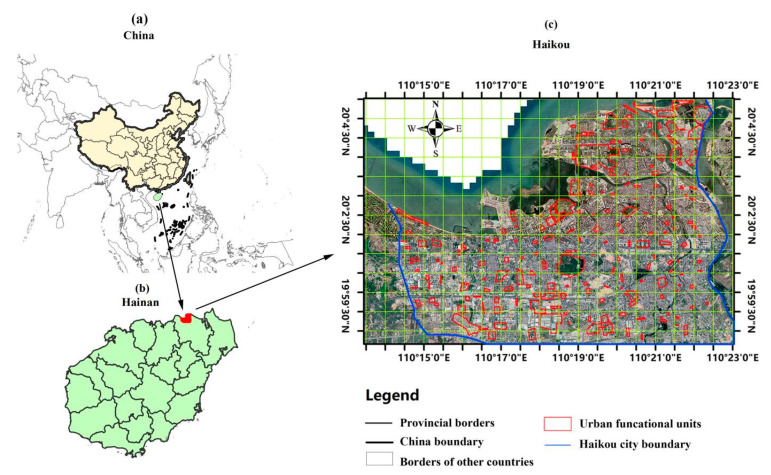
The location of Haikou; (**a**) Map of China highlighting Hainan, (**b**) map of Hainan highlighting Haikou, and (**c**) satellite imagery of Haikou.

**Figure 2 biology-11-01824-f002:**
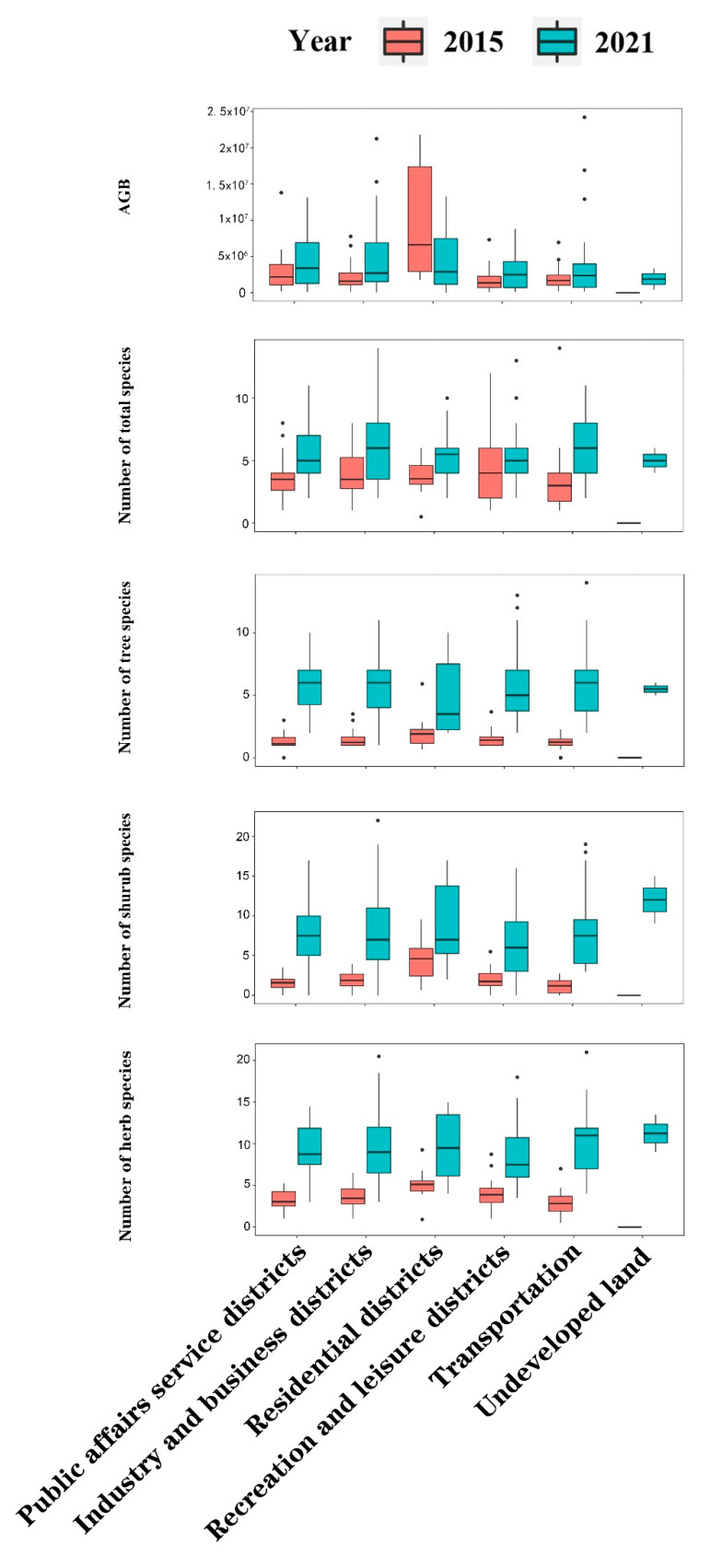
Boxplots of AGB and the number of total, tree, shrub and herb plant species in primary UFUs in 2015 and 2021. The black dots in the graph represent extreme outliers.

**Figure 3 biology-11-01824-f003:**
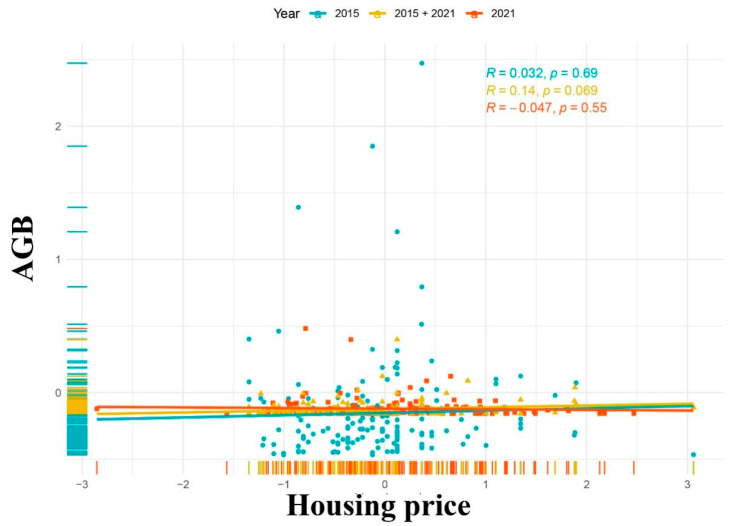
Scatterplot depicting the relationships between AGB and housing price in Haikou in 2015, 2021, and 2015 with 2021. 2015 + 2021 represents univariate analysis of the number of species in 2021 and socioeconomic factors in 2015.

**Table 1 biology-11-01824-t001:** Number of primary and secondary urban function units (UFUs) selected in Haikou.

Primary Type of UFUs	Secondary Types of UFUs	Number of UFUs
Public affairs service districts	Governmental Agencies	18
	Colleges/Universities	7
	Primary/Middle Schools	18
	Research Institutes	4
	Hospitals	12
Industry and business districts	Industry	12
	Hotels	11
	Industrial Offices	9
	Supermarkets	3
Residential districts	Low-Density Residential Areas	5
	(<6 stories)	
	High-Density Residential Areas	43
	(>6 stories)	
Recreation and leisure districts	Parks	7
	Museums	5
Transportation	Main/Secondary Roads	28
	Bus Parking	5
Undeveloped land	Wetland	3
Total		190

**Table 2 biology-11-01824-t002:** Mean and standard deviation changes of AGB and the number of total, tree, shrub and herb species in primary UFU in Haikou from 2015 to 2021.

Primary UFU Type	AGB	Number of Tree Species	Number of Shrub Species	Number of Herb Species	Number of Total Species
Public affairs service districts	2.78 × 10^7^ ± 3.27 × 10^6^	1.99 ± 0.89	4.21 ± 1.76	5.44 ± 3.19	11.93 ± 3.03
Industry and business districts	1.28 × 10^7^ ± 8.25 × 10^6^	2.06 ± 0.73	4.44 ± 1.46	5.77 ± 3.04	12.21 ± 3.7
Residential districts	1.00 × 10^7^ ± 6.50 × 10^6^	0.90 ± 0.45	4.13 ± 2.28	4.48 ± 2.91	9.51 ± 4.29
Recreation and leisure districts	−5.31 × 10^7^ ± −3.19 × 10^7^	1.89 ± 0.99	2.78 ± 1.65	−1.00 ± 1.50	9.09 ± 2.53
Transportation	2.31 × 10^7^ ± 4.44 × 10^7^	2.80 ± 1.02	4.96 ± 2.27	6.90 ± 3.93	14.67 ± 4.98
Undeveloped land	1.88 × 10^7^ ± 1.44 × 10^7^	1.72 ± 1.41	5.50 ± 0.70	3.44 ± 1.46	22.5 ± 4.5

**Table 3 biology-11-01824-t003:** Stepwise regression models with factors predicting the change of above ground biomass (AGB) from 2015 to 2021.

		2015 AGB	2021 AGB
Socioeconomic variables	Construction age (years)	-	-
	Housing price (RMB/yuan)	-	-
	Population density (inhabitants/km^2^)	-	-
	Traffic flow	0.068	-
	Distance from main road (m)	-	-
Geographical factors	Longitude	-	-
	Latitude	0.008 **	-
Greening management factors	Fertilizing frequency (times/year)	0.061	−0.089
	Maintenance frequency (times/year)	-	-
	Watering frequency (times/year)	-	-
Intercept		0.007 **	0.070
Akaike information criterion (AIC)		−314.59	−848.48
*p*-value		0.069	7.516 × 10^−5^
Adjusted R-squared		0.003	0.087

‘-’ indicates that the variable was not included in the final model. Significance codes: ‘**’ *p* < 0.01. 2015 + 2021 represents univariate analysis of the AGB in 2021 and predictors from 2015.

**Table 4 biology-11-01824-t004:** Stepwise regression model with factors predicting the number of total, tree, shrub, and herbs species in 2015 and 2021.

Predictor		2015 Tree	2021 Tree	2015 Shrubs	2021 Shrubs	2015 Herbs	2021 Herbs	2015 Total	2021 Total
		**Estimate**	**Estimate**	**Estimate**	**Estimate**	**Estimate**	**Estimate**	**Estimate**	**Estimate**
Socioeconomic variables	Construction age (years)	-	0.214	−0.482	−0.360 ***	-	-	-	−0.645 *
	Housing price (RMB/yuan)	< 0.001.	-	<−0.001	0.059	-	-	-	-
	Population density (inhabitants/km^2^)	<−0.001	0.153	< 0.001	-	-	-	-	-
	Traffic flow	−0.085 *	-	0.133 *	-	0.091	-	-	-
	Distance from main road (m)	-	-	-	0.090 *	-	−0.065	-	-
Geographical factors	Longitude	-	-	-	-	-	-	-	-
	Latitude	-	0.874	-	-	-	−0.870	-	-
Greening management factors	Fertilizing frequency (times/year)	0.102	0.276 *	-	-	−0.190	−0.256 *	0.505	-
	Maintenance frequency (times/year)	-	-	−0.126	-	-	-	-	0.417 ***
	Watering frequency (times/year)	-	-	0.524 *	-	-	-	1.494 **	-
Intercept		0.741	−0.057	−0.001	−0.014	−0.044	0.080	0.253 **	−163 *
Akaike information criterion (AIC)		−292.96	−241.22	−131.71	−292	−122.42	−312.29	−31.14	−111.38
*p*-value		<0.001	<0.001	<0.001	<0.001	<0.001	<0.001	<0.001	<0.001
Adjusted R-squared		0.765	0.765	0.269	0.823	0.313	0.837	0.137	0.234

‘-’ indicates that the variable was not included in the final model. Significance codes: ‘***’ *p* < 0.001 ‘**’ *p* < 0.01 ‘*’ *p* < 0.05. 2015 + 2021 represents univariate analysis of the number of species in 2021 and socioeconomic factors in 2015.

## Data Availability

The data presented in this study can be obtained by contacting the corresponding author.

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
