# Peer review of "Spatiotemporal Variation of Urban Plant Diversity and above Ground Biomass in Haikou, China"

_biology, 2022, doi:10.3390/biology11121824_

Round 1
Reviewer 1 Report
The paper entitled Spatiotemporal variation of urban plant diversity and above ground biomass in Haikou, China shows an important analysis of urban plant diversity and aboveground biomass considering urban functional units, socioeconomic factors and resources in the period 2015-2021, in Haikou China's Hainan Province. The paper is well written, with adequate analysis of the theme and it brings a positive response to the government investment in the green areas of Haikou. Based on this study, government actions can be focused on urban landscape planning, enabling a better quality of life for the inhabitants. I just missed in the paper an analysis of the diversity of biomass in urban functional units in graphical form. This was just mentioned in the discussion. You could do a non-metric multidimensional scaling analysis to analyze these variables. Below are some suggestions for improving the paper.
Line 73: ...and spatiotemporal variations...
Line 87:..., the total area...
Figure 1: The map is too small. Enlarge map dimensions and include Asia’s map.
Lines 176 a 177: “According to our field surveys, we found 667 plant species belonging to 99 families in 2015 and 898 plant species belonging to 151 families.”
Was it 898 plant species in 2021?
Table 3: Pattern the dots after the numbers.
Line 262: “Air pollution is a major problem arising mainly from..”
Place as a next paragraph.
Lines 370-371: UFUs are ideal models for analyzing the components of urban...
Author Response
Dear Reviewer,
We write all the revised responses in Word, please see the attachment.
Best Regards,
Haili

Reviewer 2 Report
I've finished reviewing the paper titled "Spatiotemporal variation of urban plant diversity and aboveground biomass in Haikou, China”. This work has strong findings and is original, perceptive, and flawlessly written. The topic of the paper is relevant and current, and its structure is appropriate and consistent with that of the journal (including text, tables, figures and references). The organization and content of each section are suitable. Researchers examined the relationships between socioeconomic characteristics and changes in urban plant variety in this study. They looked at the connections between aboveground biomass, the quantity of trees, shrubs, and herbs, and socioeconomic factors across functional units of urban areas. According to the authors' conclusions, greater private, commercial, and governmental initiatives are needed to boost aboveground biomass in Haikou.
I think this manuscript should be published in the journal Biology after the authors fix some of the minor issues noted below because this is an interesting topic that deserves attention.
Lines 40 – 43: Please merge these two sentences into one. Generally, too short sentences are used in the Introduction section.
Lines 61 – 62: The sentence „Different management regimes are known to alter UPD and composition [11]“ is unclear. Composition of what?
Line 73: This should be one sentence, not two: To better understand the driving mechanisms and spatial variations of UPD and AGB in coastal urbanized areas. We took Haikou's urban green space as a research case... Please replace the period with a comma.
Line 87: 3145.93 km2 – correct to km2 (superscript)
Lines 124 – 125: ....cultivated plants are those that require human disturbance... Change to: human maintenance.
Do you have information on subspontaneous plant species - especially non-native invasive species? If you have, please add text to the Introduction about those species.
Line 143: You shouldn't start a paragraph with a subheading: „Survey of socioeconomic variables“.
Line 150: Is it possible to display data and results on allochthonous (invasive plant species) and the difference in their presence in 2015. and 2021.?
Line 324: The sentence: „Mowing lawns to increase the herbage area“ is a continuation of the previous sentence and should be connected to it.
Lines 362 – 382 (Conclusions): I recommend that the Conclusions chapter be shorter and more concise, without this introductory part where it repeats what has already been said in other chapters. I recommend to immediately move on to the conclusions derived from the results of the work.
Author Response
Dear Reviewer,
We write all the revised responses in Word, please see the attachment.
Best regards,
Haili
